# Machine Learning Prediction and Phyloanatomic Modeling of Viral Neuroadaptive Signatures in the Macaque Model of HIV-Mediated Neuropathology

Andrea S. Ramirez-Mata,[a,b] David Ostrov,[b] Marco Salemi,[a,b] Simone Marini,[a,b,c] Brittany Rife Magalis[a,b]

aEmerging Pathogens Institute, University of Florida, Gainesville, Florida, USA
bDepartment of Pathology, Immunology and Laboratory Medicine, University of Florida, Gainesville, Florida, USA
cDepartment of Epidemiology, University of Florida, Gainesville, Florida, USA

**ABSTRACT** In human immunodeficiency virus (HIV) infection, virus replication in and adaptation to the central nervous system (CNS) can result in neurocognitive deficits in approximately 25% of patients with unsuppressed viremia. While no single viral mutation can be agreed upon as distinguishing the neuroadapted population, earlier studies have demonstrated that a machine learning (ML) approach could be applied to identify a collection of mutational signatures within the virus envelope glycoprotein (Gp120) predictive of disease. The S[imian]IV-infected macaque is a widely used animal model of HIV neuropathology, allowing in-depth tissue sampling infeasible for human patients. Yet, translational impact of the ML approach within the context of the macaque model has not been tested, much less the capacity for early prediction in other, noninvasive tissues. We applied the previously described ML approach to prediction of SIV-mediated encephalitis (SIVE) using *gp120* sequences obtained from the CNS of animals with and without SIVE with 97% accuracy. The presence of SIVE signatures at earlier time points of infection in non-CNS tissues indicated these signatures cannot be used in a clinical setting; however, combined with protein structural mapping and statistical phylogenetic inference, results revealed common denominators associated with these signatures, including 2-acetamido-2-deoxy-beta-D-glucopyranose structural interactions and high rate of alveolar macrophage (AM) infection. AMs were also determined to be the phyloanatomic source of cranial virus in SIVE animals, but not in animals that did not develop SIVE, implicating a role for these cells in the evolution of the signatures identified as predictive of both HIV and SIV neuropathology.

**IMPORTANCE** HIV-associated neurocognitive disorders remain prevalent among persons living with HIV (PLWH) owing to our limited understanding of the contributing viral mechanisms and ability to predict disease onset. We have expanded on a machine learning method previously used on HIV genetic sequence data to predict neurocognitive impairment in PLWH to the more extensively sampled SIV-infected macaque model in order to (i) determine the translatability of the animal model and (ii) more accurately characterize the predictive capacity of the method. We identified eight amino acid and/or biochemical signatures in the SIV envelope glycoprotein, the most predominant of which demonstrated the potential for aminoglycan interaction characteristic of previously identified HIV signatures. These signatures were not isolated to specific points in time or to the central nervous system, limiting their use as an accurate clinical predictor of neuropathogenesis; however, statistical phylogenetic and signature pattern analyses implicate the lungs as a key player in the emergence of neuroadapted viruses.

**KEYWORDS** SIV, HIV, machine learning, neuropathology, neuroadaptation, envelope, neuroAIDS, phyloanatomy, phylogenetic

Address correspondence to Brittany Rife Magalis, brittany.rife@ufl.edu, or Simone Marini, simone.marini@ufl.edu.

The authors declare no conflict of interest.

**H**uman immunodeficiency virus (HIV) is capable of infecting the central nervous system (CNS) and significantly damaging neuronal cells, ultimately leading to neurocognitive impairment. Clinical manifestations of HIV-associated neurocognitive disease (HAND) can include decreased attention and concentration, memory loss, reduced psychomotor ability and executive function, and tremors; with time, these manifestations can progress to dementia (1). Despite the success of antiretroviral therapies (ART) in countries where they are widely available, HAND continues to persist among persons living with HIV (PLWH). Though more commonly observed among PLWH that have progressed to AIDS (2), not all PLWH (approximately 25%) are diagnosed (3), suggesting a distinct viral-, immune-, or potentially drug-mediated mechanism associated with disease pathogenesis that has yet to be identified (4, 5).

For infection of the brain to occur, the virus must enter by way of the highly selective blood-brain barrier and replicate in a subset of cells (perivascular macrophages and microglia) that are unique to this tissue (6). The CNS thus constitutes a distinctive microenvironment imparting measurable selective pressure(s) that can give rise to tissue-adapted viral variants (7–10). Multiple studies have explored further aspects of this hypothesis and concluded that there exist HIV-1 envelope (*Env*) amino acid patterns, or signatures, unique to viral sequences found in the CNS (7, 11–14). Such individual signatures, however, vary depending on the study, rendering results inconclusive as to a single, critical mutation and/or mutational region associated with neuroadaptation. The controversy surrounding the existence of a neuroadaptive signature may be explained by failing to consider the combined effect of multiple, noncontiguous amino acid residues and their biochemical properties (15).

Machine learning (ML) approaches have been more recently applied in genomic analysis to model the correlation between disease variants and clinical outcomes, capitalizing on the presence of multiple, combinatorial viral factors (16–18). Holman and Gabuzda (15) developed an ML pipeline paving the way for the use of HIV variants in predicting neuropathological outcome. Their study identified five statistically significant amino acid signatures that were capable of predicting with 75% accuracy the outcome of HIV-associated dementia (HAD). However, this study was limited to analysis of a small portion of the gene—the V3 loop and surrounding C2 and C3 regions of the envelope glycoprotein (Gp120)—and the position of the signatures within the linear sequence (i.e., no protein structure context). Ogishi and Yotsuyanagi (19) also generated a ML prediction model for HAND from a comprehensive data set, similarly to Holman and Gabuzda (15) considering only the C2-C3 region, in which they used an iterative ML and stepwise feature reduction. With this methodology, they obtained accuracy of 100% in a hold-out testing subdata set and 95% using the entire data set, reporting that as few as three genetic features were sufficient to predict HAND status regardless of sampled tissue. Owing to more general limitations of the use of human subjects, no conclusions could be drawn regarding how early this prediction could be made in time; nor whether these signatures could be found in non-CNS tissues that comprise the potential origins for neurovirulence and thus targets for prevention of CNS invasion (20).

The S[imian]IV-infected macaque model can be used to recapitulate neuropathogenesis associated with HAND and provides a unique opportunity to acquire additional tissue samples at numerous time points throughout the course of infection (21, 22). Approximately 30% of SIV-infected Rhesus macaques that naturally progress to simian AIDS (i.e., without immune modulation) develop the pathological hallmark of neuroAIDS —SIV-mediated encephalitis (SIVE)—in 2 to 3 years. For increased incidence and/or more rapid progression to SIVE, models have been developed using neurovirulent viral strains or (23) antibody-mediated depletion of CD8$^+$ lymphocytes (24). In this study, we explore the existence of SIV genetic and/or biochemical signatures across these different animal models that are predictive of HIV/SIV-mediated neuropathology. In addition to characterizing the translational applicability of results from earlier studies in HIV-infected individuals using protein structural data, we significantly expanded on these studies by (i) using

**TABLE 1** Summary of macaques included in the training, validation and application data sets

| Data use | | SIVE | SIVnoE | Total |
|---|---|---|---|---|
| Training and validation | Publications | Matsuda et al (23); Rife et al. (7) | Perez et al. (25); Rife et al (7) | |
| | Inoculating virus | SIVsm804E-CL757, SIVmac251 | SIVmac251 | |
| | Animal species | Rhesus macaque | Rhesus macaque | |
| | Animal model(s) | Naturally progressing[a], CD8-depleted | Naturally progressing[a], CD8-depleted | |
| | Animal number | 7 | 8 | 15 |
| | Tissue types | Brain, meninges | Brain, spinal cord | |
| | Sequence no. | 334 | 219 | 553 |
| | Sequence origination | TOPO TA cloning, SGA[b] | PCR purification, SGA[b] | |
| | cART | None | 2 | 2 |
| Application | Publications | Rife et al. (7) | Rife et al. (7), unpublished | |
| | Inoculating virus | SIVmac251 | SIVmac251 | |
| | Animal species | Rhesus macaque | Rhesus macaque | |
| | Animal model(s) | CD8-depleted, naturally progressing | CD8-depleted, naturally progressing | |
| | Animal number | 4 | 9 | 13 |
| | Sequence no. | 993 | 2,779 | 3,772 |
| | Tissue types | CD3+, CD14+, bone marrow, BAL[c] fluid, plasma, lymph node | CD3+, CD14+, bone marrow, BAL[c] fluid, plasma, lymph node | |
| | Sequence origination | SGA[b] | SGA[b] | |
| | cART | None | 1 | 1 |

[a]Natural progression refers to the lack of immune modulation (e.g., CD8+ lymphocyte depletion).
[b]SGA: single-genome amplification.
[c]bronchoalveolar lavage.

the full envelope (*gp120*) region for model training and (ii) applying the learned signatures to additional SIV sequences from a variety of tissues and time points, therefore exploring SIVE intrahost heterogeneity in terms of time and tissue trajectories.

## RESULTS

The main objective of this study was to identify SIV protein signatures associated with, and predictive of, AIDS-related neuropathology in multiple macaque models. Viral Gp120 sequences used in model training were extracted from the CNS of animals originating from multiple cohorts comprised of different macaque models of disease progression, (Table 1), including infection with a neurovirulent strain (23) and CD8+ lymphocyte depletion (7). Each animal was histopathologically diagnosed at necropsy as with (SIVE) or without SIVE (SIVnoE). We estimated the performance of the model by leave-one-animal-out cross-validation, iteratively removing all sequences from a single animal and using them as test data while considering the remaining sequences and animals as training. For each test fold of the cross-validation we classified both the single sequence and the animal as a whole, where animals were diagnosed with SIVE if the majority (>50%) of sequences in that animal were classified as such. The model outputs rules comprised of a set of requirements referred to as amino acid signatures that sequences must follow to be classified as SIVE or SIVnoE. Our final model was then applied to the sequences from non-CNS tissues at different collection time points during infection (application data set) (Table 1) to evaluate spatiotemporal patterns of signature presence.

**Machine learning to extract genetic signatures.** We applied a machine learning approach like that used by Holman and Gabuzda (15) to SIV Gp120 amino acid sequences obtained from the CNS of animals with or without SIVE (Table 1). We proceeded according to the following three main steps: (i) feature/attribute selection (independent for each fold in cross-validation), (ii) classification and model selection, and (iii) prediction assessment, wherein steps 1 and 2 were used in model training, and step 3 served in the evaluation of the predictive performance (26). A schematic summary of the implemented pipeline is illustrated in Fig. 1. A detailed description of the approach can also be found in Materials and Methods. Scripts used in the training, validation, and application of the model are available at https://github.com/salemilab/neuroSIVirulence.

**Macaques as independent samples.** Phylogenetic tree reconstruction was performed for aligned SIV Gp120 amino acid and nucleotide sequences obtained from a

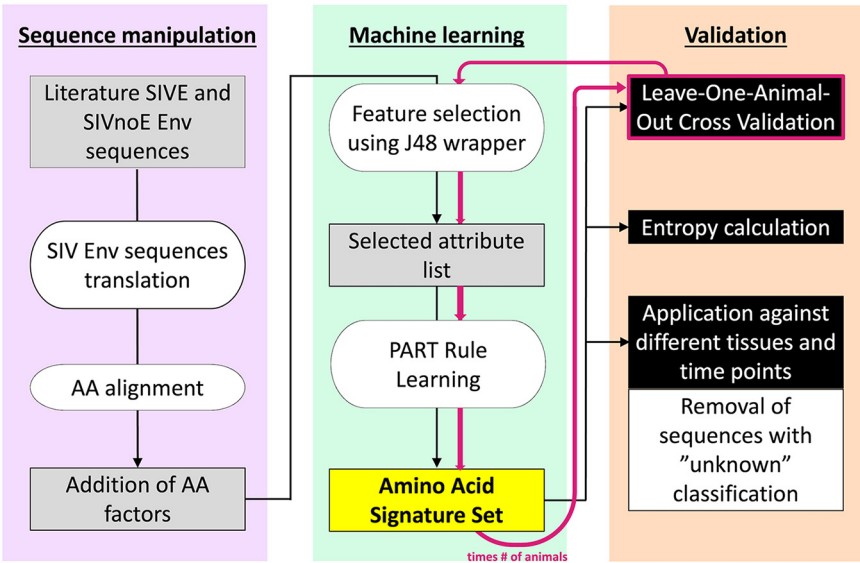

**FIG 1** Schematic of the machine learning (ML) pipeline used in the identification and validation of viral amino acid signatures associated with SIVE. The first step corresponds to sequence manipulation wherein DNA sequences are translated and aligned, followed by annotation of each amino acid according to four biochemical features: molecular size, electrostatic charge, polarity, and contribution to secondary structure (27). Feature selection is then performed, filtering on the most informative attributes. The final ML model is obtained from these selected attributes, and amino acid signature set(s) (or rule[s]) is/are obtained. In order to evaluate the model, a leave-one-animal-out cross-validation was performed. The final model was then applied to sequences from remaining tissues and time points (if available) from the same animals, as well as additional animals.

variety of tissues and time points (when available) from animals diagnosed as SIVE or SIVnoE. Phylogenetic clustering according to SIVE status or cohort was not observed (Fig. 2A and B), indicating the absence of cohort-specific selective pressure and significant convergent evolution of a neuroadaptive genotype, like Holman and Gabuzda (15). Whereas macaque-specific clustering of amino acid sequences was not observed for necropsy samples used in training and validation, corresponding nucleotide sequences clustered significantly (bootstrap support ≥ 90%) according to animal (Fig. 2B). This finding is consistent with the presence of immune-mediated convergent evolution at the protein level (28) but is also indicative of host-specific genetic drift of the virus population and, thus, allowing for each animal to be treated independently within the model.

**Predictive genetic signatures of SIVE status and model performance.** Individual amino acids within necropsy sequences originating in the CNS (including brain, meninges, and spinal cord) were assigned standardized values for the following biochemical properties, as described in Atchley et al. (27)—polarity, electrostatic charge, molecular size, and secondary structure (also used by Holman and Gabuzda [15]). The ML prediction model was trained using the projective adaptive resonance theory (PART) rule-learning algorithm, resulting in a collection of rules, each comprised of set of amino acid signatures. Performance of the model was evaluated using leave-one-animal-out cross-validation (Fig. 1).

The final model extracted 15 rules comprising signature sets predictive of SIVE. Sequences not classified as SIVE according to signature presence were considered to fall under the category of SIVnoE. Following feature selection, amino acid identity, polarity, and molecular size were considered the most informative for SIVE prediction. An example of these and other relevant features is represented in Fig. 3A. Individual rules containing a subset of signatures from a larger rule (e.g., rule 1_01) were considered nested, forming a network, or rule group (Fig. 3B). SIVE rule groups were ultimately comprised of eight amino acid sites within the alignment (Fig. 3C). Electrostatic charge and contribution to secondary structure were not considered informative for these data, in contrast with Holman and Gabuzda (15), who reported all four biochemical properties as important

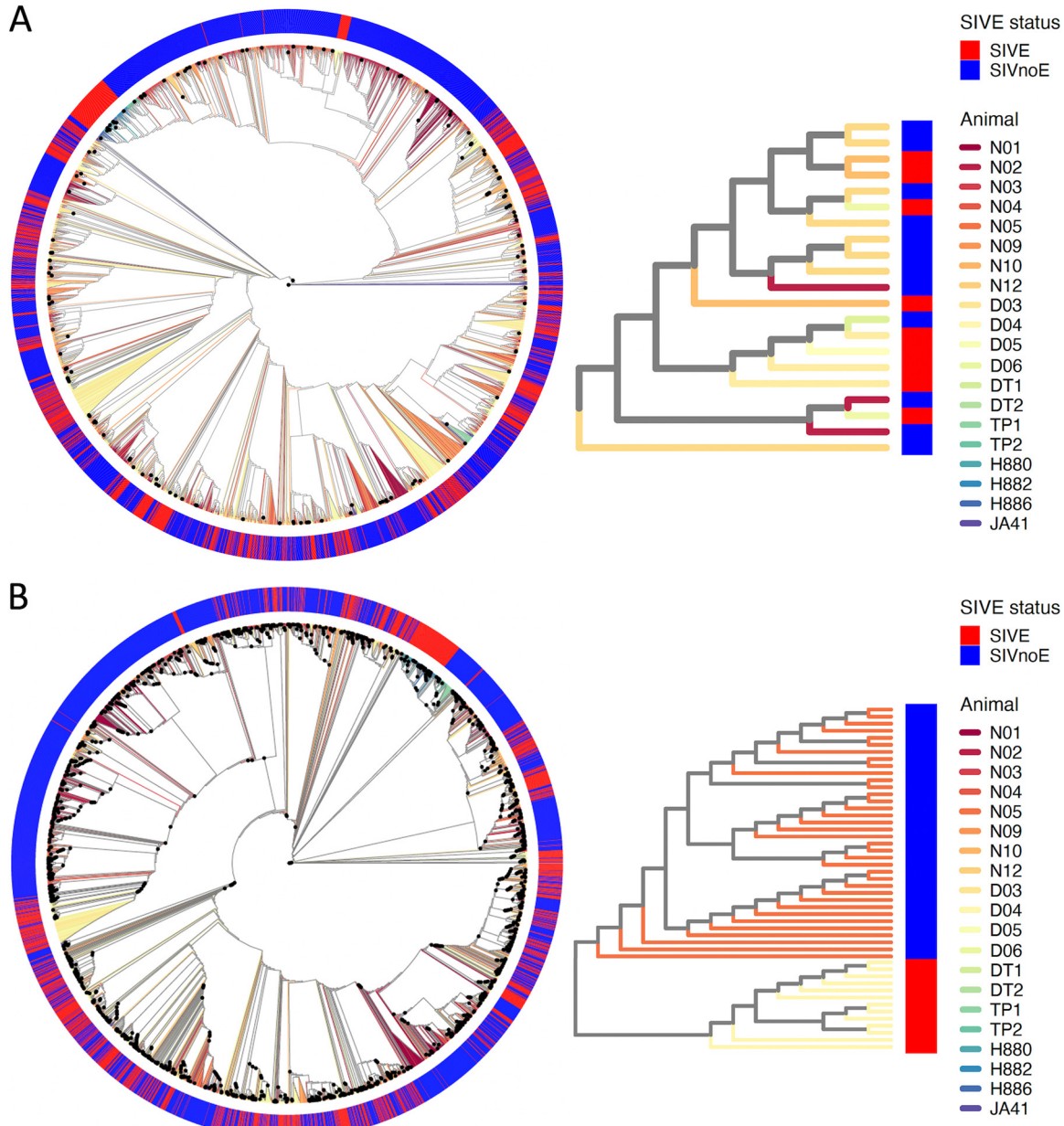

**FIG 2** Maximum likelihood phylogenetic trees reconstructed from SIV envelope glycoprotein amino acid and nucleotide sequences from animals diagnosed with (SIVE) or without (SIVnoE) SIV-associated encephalitis. (A) Phylogenetic tree (left) reconstructed with all amino acid sequences used in model training, validation, and application. Each animal in the tree is represented by a different branch color, as well as color within the corresponding heatmap representing SIVE status (legend right). Bootstrap support values ≥90% are represented by black dots at internal nodes within the tree. An inset (right) of the full tree demonstrates the absence of amino acid sequence clustering according to SIVE status, cohort, and animal. (B) Phylogenetic tree (left) reconstructed with all nucleotide sequences used in model training, validation, and application. An inset (right) of the full tree demonstrates animal-specific nucleotide sequence clustering for samples collected at necropsy.

features. Following cross-validation, these rules demonstrated a balanced accuracy of 97% (Table 2), with signature set 1_05 having correctly classified the greatest number of true SIVE animals (6) and CNS sequences from SIVE animals (206) (Table 3). All cohorts (and animals) were represented in the total rule set (Table S2). Precision and recall (as well as F1 score) were also >95%, indicating the model was both sensitive and specific to the detection of SIVE signatures and an overall adequate model for SIVE prediction.

**Mutational entropy, coevolution, and selection among SIVE signature sites.** Amino acid site 344 was the most frequently represented (7 of 15 rules), for which amino acid

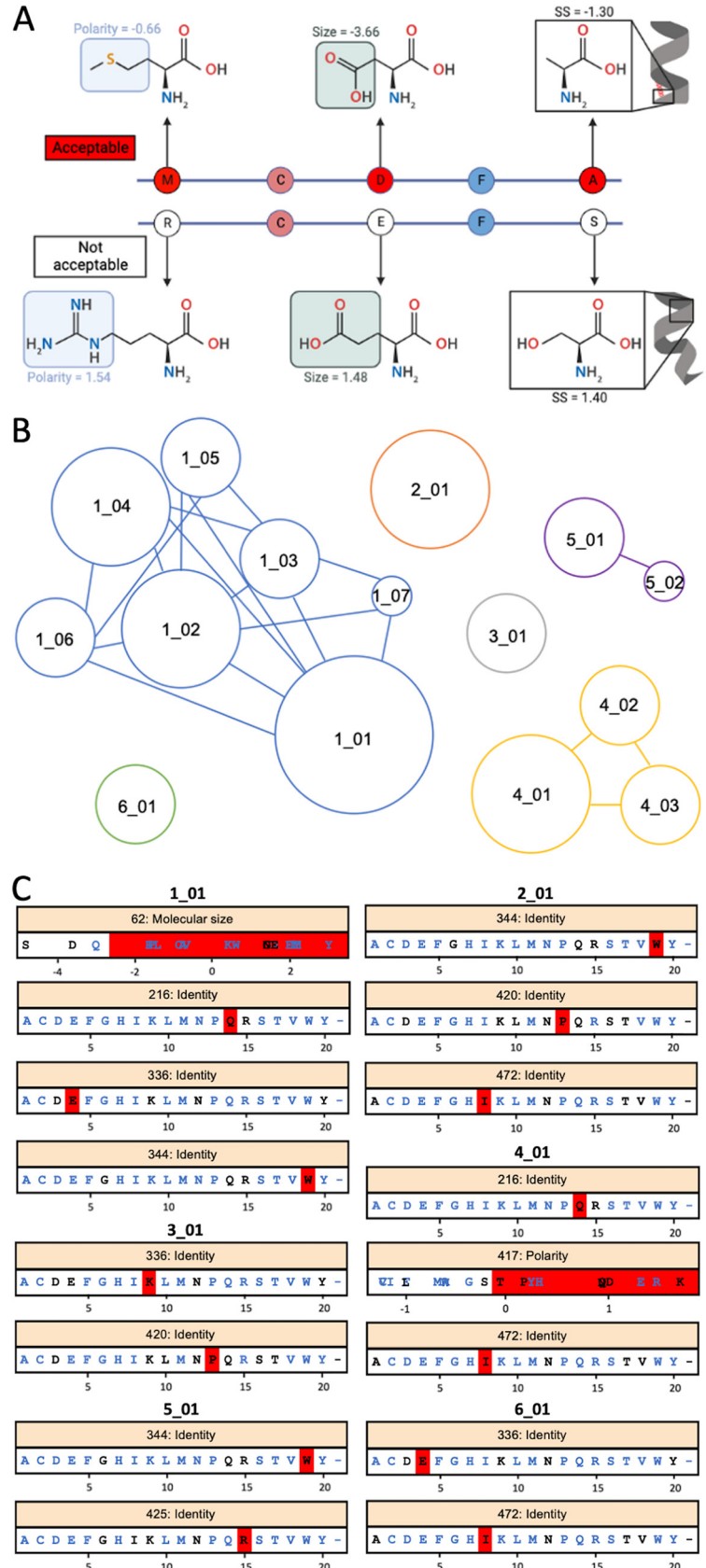

FIG 3 SIVE rules based on the machine learning model used for necropsy-sampled CNS sequences from SIVE and SIVnoE animals. (A) Representation of acceptable (top) and nonacceptable (bottom) biochemical

**TABLE 2** Performance statistics of the machine learning model generated with the validation data set consisting of necropsy-sampled CNS sequences from SIVE and SIVnoE animals[a]

| Statistic | Validation set per sequence | Validation set per animal[a] |
|---|---|---|
| Precision | 0.96 | 0.90 |
| Recall | 0.97 | 0.86 |
| Balanced accuracy | 0.97 | 0.89 |
| F1 score | 0.97 | 0.86 |

[a]Animal classification based on majority SIVE sequence classification.

identity was restricted to tryptophan for all rules for SIVE prediction. This site, along with the second-most-frequent signature site (336 in 6 of 15 rules), is located within the V3 region of Gp120 (Fig. 4A). Variable region V3 was second in terms of total signature site number only to region V4, containing three relatively neighboring sites—417, 420, and 425. In general, however, rules were not restricted to one particular region within the protein. Results resembled those of the study by Holman and Gabuzda (15), which, although limited to the C2-C3 region, uncovered a prominent role for V3. The increased presence of signature sites within this known variable region could not be explained entirely by mutational bias in this previous study, as positions included in signatures exhibited a wide range of mutational entropy (15). Consistent with this finding, we observed a similar wide range in entropy values across both signature and nonsignature sites (Fig. 4B), suggesting sites experiencing elevated evolutionary rates were not restricted to neuroadaptive function and that the model was not restricted to mutable sites. However, signature sites experienced approximately 8-fold greater entropy on average than sites not designated signature sites, which was considered significant ($P < 0.01$; Fig. S1).

To evaluate further the potential mutational bias of the PART algorithm, we performed a Bayesian graphical model (BGM) analysis of pairwise mutational associations (i.e., coevolution of sites) between amino acid positions. This approach was specifically used to determine if the tendency for higher-entropy sites to be included in SIVE signature sets could be explained by epistatic interactions between signature sites, represented by frequent covariation of sites along the same branch within the phylogeny and often observed during adapting to new environments (29). A total of 147 sites were considered to be coevolving, including four of the eight signature-associated sites (Fig. 4C). None of these four sites were coevolving with each other, suggesting evolutionarily independence of each signature site. The observance of coevolution of signature sites with nonsignature sites suggests residues involved in neuroadaptation likely perform additional functions (e.g., protein folding) on which other nonsignature residues may be conditionally dependent.

Alternative to epistasis, molecular adaptation at the level of an individual residue often requires flexibility in terms of amino acid change, represented as elevated mutational entropy, and can be driven by positive selection. Selection experienced by individual amino acids can be identified, and quantified, through the estimation of underlying relative nonsynonymous (*dN*) and synonymous (*dS*) codon substitution rates within the nucleotide data. Despite the relatively wide range of mutational entropy between the two signature sites, selection pressure experienced by sites required for neurovirulence may occur only transiently during the course of infection in the midst of dynamic immune responses (30) and target cell distributions (31), resulting in fluctuations in *dN* over time. We, therefore, applied the mixed effects model of evolution (MEME) to the nucleotide sequence data, designed to detect both pervasive and episodic selection (32). Using this approach, a total

**FIG 3** Legend (Continued)

properties — polarity, molecular size, and contribution to secondary structure (SS) — for example, amino acid sites within two linear protein sequences. (B) Rule group networks, demonstrating rules connected via overlap of at least one amino acid site. Size of rule node is proportional to the number of associated signatures. Colors represent individual rule groups. (C) Acceptable amino acid identities or feature values for each major rule. For each position, amino acids observed at that position are colored in black, positions not observed are in blue. The red bar indicates the range of acceptable values in that signature, according to the obtained rules using the PART algorithm. Positions given are based on SIVmac251 reference sequence KU892415.1.

**TABLE 3** SIVE classification in animals diagnosed with SIVE

| Signature set | No. of sequences ($n = 553$) | No. of animals ($n = 7$) |
|---|---|---|
| 1_01 | 98 | 3 |
| 1_02 | 100 | 3 |
| 1_03 | 163 | 5 |
| 1_04 | 135 | 4 |
| 1_05 | 206 | 6 |
| 1_06 | 129 | 3 |
| 1_07 | 13 | 0 |
| 2_01 | 17 | 0 |
| 3_01 | 13 | 0 |
| 4_01 | 151 | 3 |
| 4_02 | 170 | 4 |
| 4_03 | 151 | 3 |
| 5_01 | 67 | 2 |
| 5_02 | 67 | 2 |

of 127 amino acid sites were identified as experiencing significant positive selective pressure ($P < 0.10$). All signature sites predictive of SIVE were considered to have experienced significant positive selection pressure, though not restricted to SIVE animals (Fig. 5A). Whereas positive selection of signature sites was considered approximately 2-fold greater than that of nonsignature sites, selection for signature sites was only observed on average across 1.8% of branches (Fig. S2), indicating selection was largely episodic and potentially explaining why positive selection for these sites was largely not observed among the SIVsm804E-CL757-infected animals sampled only at necropsy (Fig. 5B). Of the eight sites, highly prevalent V3 signature site 344 and less prevalent V4 signature site 420 were the only sites observed among both SIVmac251- and SIVsm804E-CL757-infected cohorts, among which site 344 was specific to the CD8-depleted animals only and site 420 to both CD8-depleted and nondepleted animals. Assuming signature sites not observed in the SIVsm804E-CL757 cohort could be required to maintain fitness in animals directly infected with a neurovirulent clone, analysis of purifying selection was also undertaken using a fast unconstrained Bayesian approximation (FUBAR) (33); however, synonymous rate variation was not considered significantly greater ($P < 0.05$) than that of nonsynonymous substitutions for these sites in these animals.

**Spatiotemporal distribution of SIVE-associated amino acid signature sets.** The learned model was next applied to sequence data from additional sampled tissues and time points from an expanded animal cohort to gain better insight into how early viral-mediated neuropathology can be predicted and the invasiveness of the sampling procedure required for reliable SIVE prediction. All remaining tissues and time points for animals described in Rife et al. (7) were used in this application, as well as samples from an additional animal receiving antiretroviral therapy (Table 1, Fig. 6A). The observance of SIVE signature sets in tissues outside the CNS and at earlier time points, at times as high as 100% of sequences (Fig. 6B) was promising, as it demonstrated the potential for prediction of neuropathology without invasive collection of CNS samples and potentially at a time point earlier than pathological onset that, if early enough, may provide sufficient time for therapy-mediated prevention. However, SIVnoE animals also exhibited the presence of SIVE signature sets at earlier time points (Fig. 6B). A high rate of SIVE classification among early sampled sequences in both cohorts, particularly for rule groups 1 and 5, might not necessarily be a misclassification of the model, but rather an indication of the presence of SIVE signatures in the infecting viral swarm (SIVmac251), which was originally obtained from monkeys at the time of necropsy (34, 35). It is possible that this infecting population/quasispecies may harbor neuroadaptive signatures that are not beneficial at the time of, or immediately following, infection/transmission.

A high degree of presence of SIVE signatures in non-CNS tissues was not anticipated, considering CNS sequences have historically been phylogenetically distinct

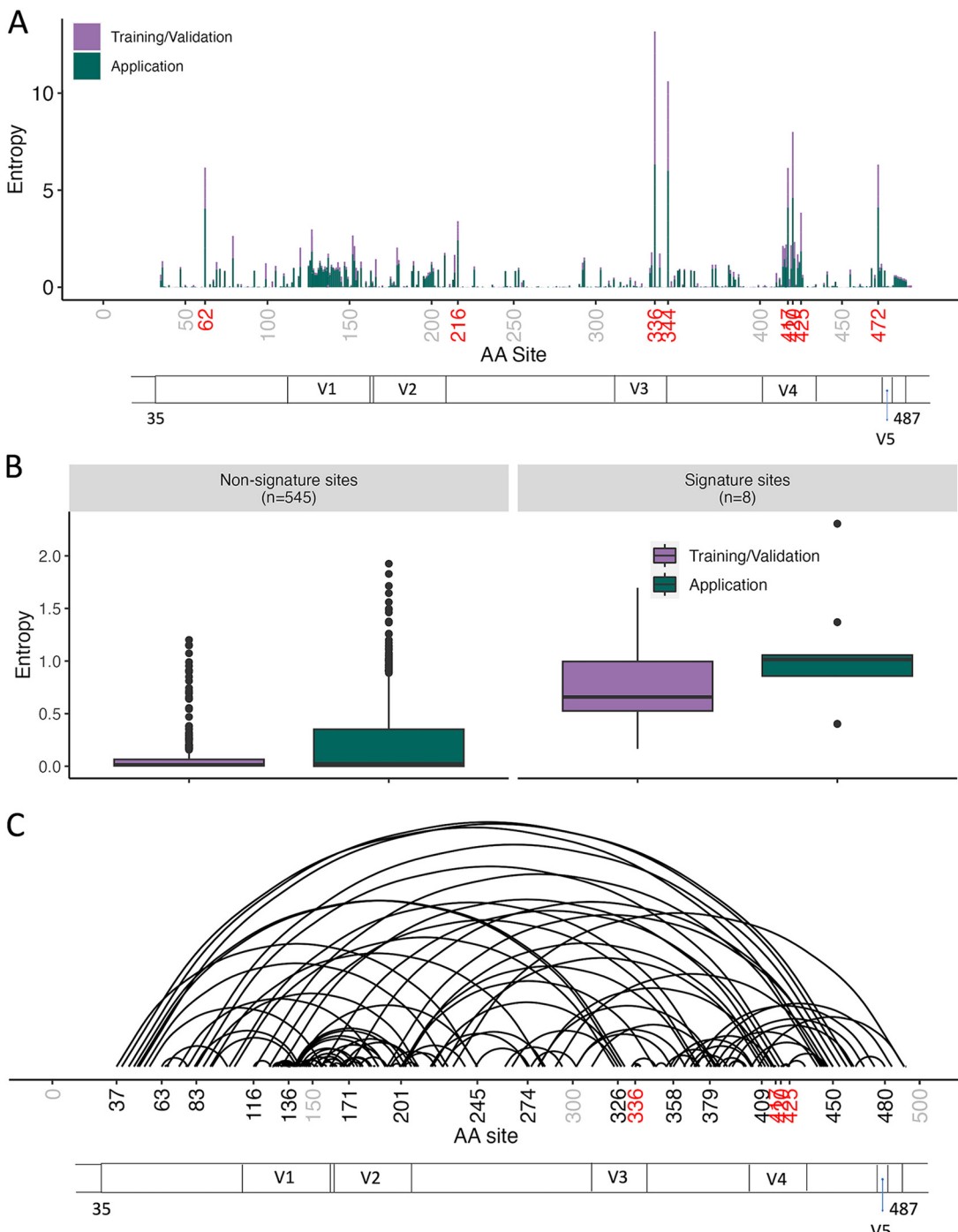

**FIG 4** Mutational entropy and coevolution across Gp120 amino acid sites. (A) Shannon entropy values (y axis) for each amino acid in training (green) and testing (purple) data sets. Signature-associated amino acid positions (x axis) are highlighted in red. (B) Entropy value distributions (y axis) among amino acid positions according to SIVE signature association and data set assignment (x axis). (C) Conditional dependencies obtained from the Bayesian graphical model (BGM) for SIV Gp120 amino acid sequence mutations. Arcs represent the association of two amino acid positions (x axis) exhibiting significant covariation ($P \geq 50\%$). Sites identified using the BGM model are displayed in black (not all sites present), whereas signature sites are highlighted in red. Gray site numbers indicate positional markers only (no dependency observed). SIV Gp120 constant (C) and variable (V) regions are also shown along the x axis of panels A and C. Position numbers are based on the SIVmac251 reference sequence (KU892415.1, protein ID AMX21539.1).

from sequences collected outside the CNS (8, 36–38). The presence of shared genetic signatures across these otherwise compartmentalized sequences indicates that, in the face of natural evolution of the virus in the CNS, some specific protein property required for CNS entry and/or replication is not entirely unique to the CNS. The ability of the virus

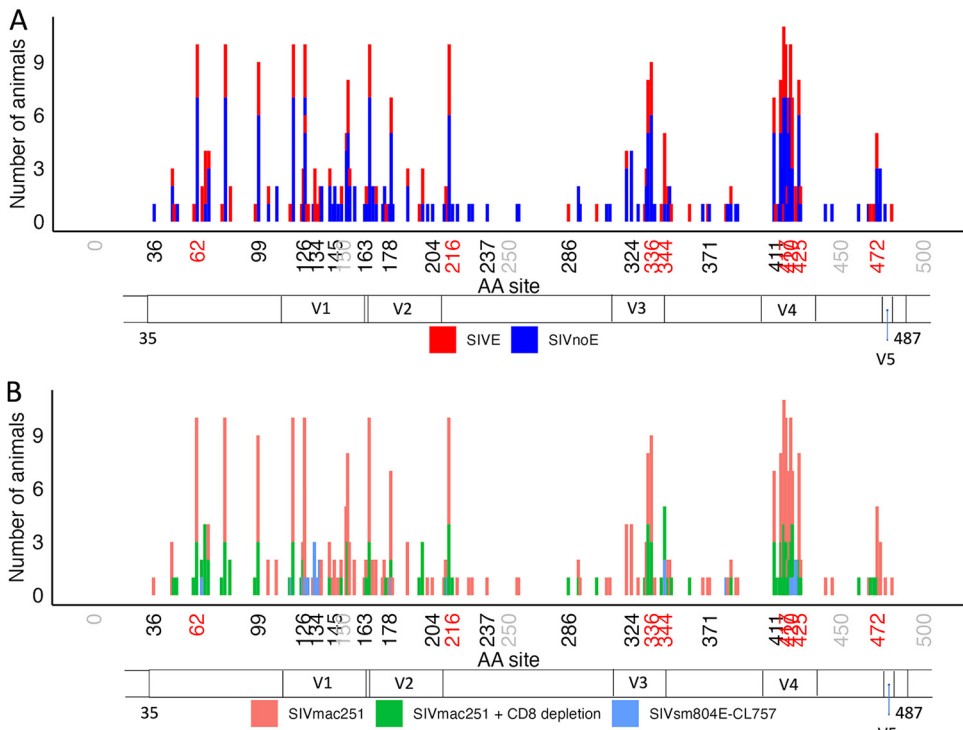

**FIG 5** SIV Gp120 amino acid sites considered to have experienced episodic or pervasive positive selection across macaque cohorts used in model training, validation, and application. Amino acid sites (x axis) determined to be under positive selection according to the MEME model of episodic selection (32), represented in terms of frequency among SIVE and SIVnoE animals (A) or among cohort (B). Gray site numbers indicate positional markers only (not under selection), black numbers indicate positively selected nonsignature sites, and red numbers represent positively selected sites belonging to SIVE prediction rules. SIV Gp120 constant (C) and variable (V) regions are also shown along the x axis of both panels. Position numbers are based on the SIVmac251 reference sequence (KU892415.1, protein ID AMX21539.1).

to adapt to new cell types, for example, is well known. The development of macrophage tropism is particularly well-characterized and has been linked previously to neuropathology (39, 40). As the primary cell types infected within the CNS are perivascular macrophages and microglia (6), it is not unreasonable to rationalize that cellular tropism is a necessary phenotype for CNS infection. Indeed, 100% of lung macrophages comprising the bronchoalveolar lavage (BAL) fluid were classified as SIVE in SIVE-diagnosed animals during late-stage infection using this ML model. As this population was also classified as SIVE at high levels in SIVnoE animals at earlier time points (Fig. 6B), we posit that macrophage tropism is not the only requirement for neuroadaptation. Moreover, macrophage tropism has been reported in a variety of individuals, regardless of neuropathology (41), suggesting that macrophage-tropic variants might be established and replicating in macrophages before neurological pathology becomes apparent.

**Structural similarities between SIV and HIV Gp120 signatures.** Next, we wanted to examine the extent to which the results of Holman and Gabuzda (15) in PLWH were translatable to the SIV-macaque model. Like HIV-1, SIV makes use of envelope glycoproteins to bind to cellular surface receptors and infect their target cells. Although macaque SIV and HIV-1 share only ~35% sequence identity in their envelope glycoproteins, they exhibit higher similarity in other, more functional aspects (e.g., 70% similarity in the location of disulfide bonds), and their molecular architectures are highly comparable (42). Hence, direct translation of amino acid position, or even biochemical property, within the neighboring area between viruses was not anticipated. Instead, we used published Gp120 protein structures for HIV-1 (PDB 2NY3) (43) and SIV (PDB 3JCC) (44), highlighting the signature-associated amino acid positions obtained from Holman and Gabuzda (15) and herein, respectively, in order to look for structural clues in terms of functional

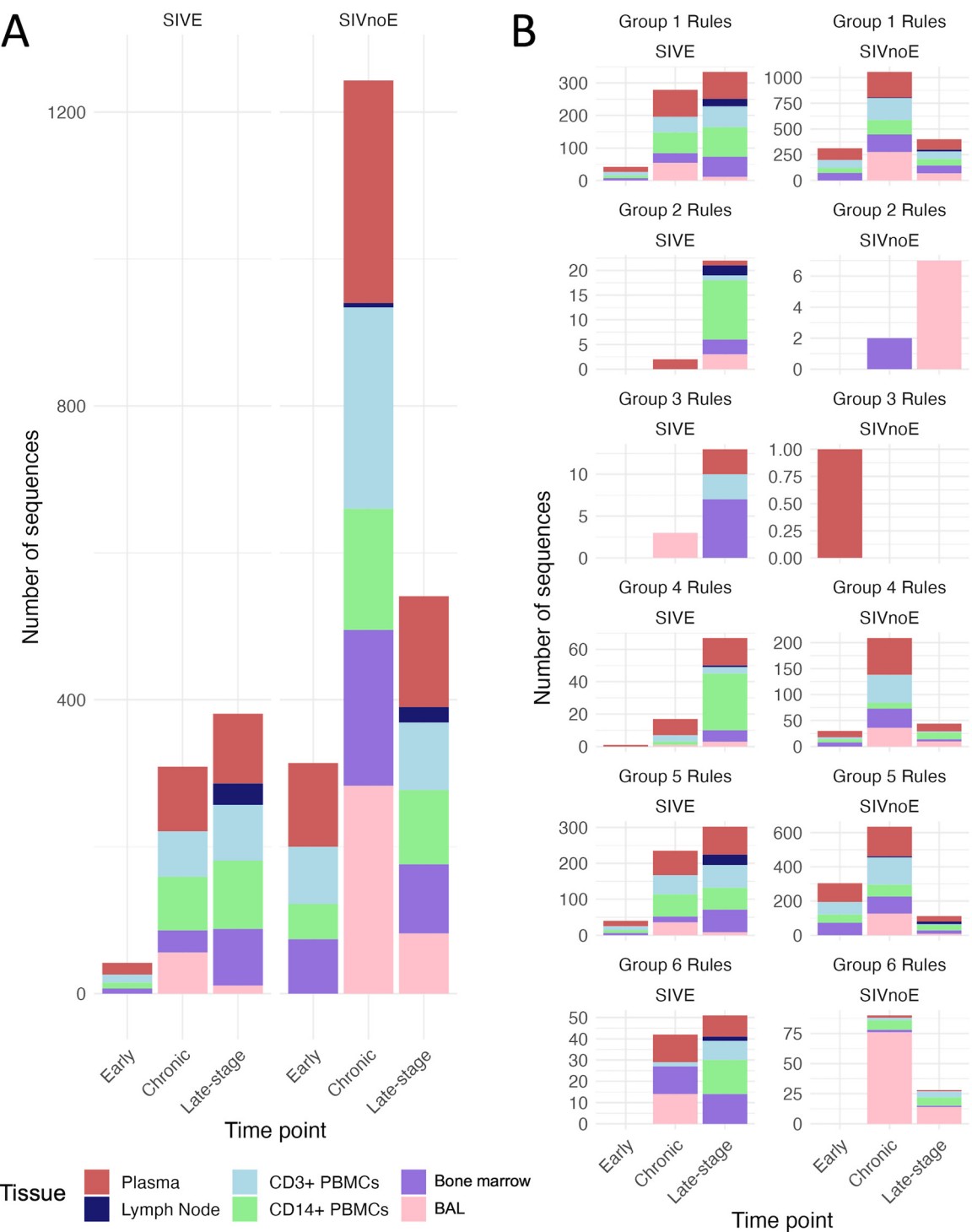

**FIG 6** Classification of sequences obtained from tissues over the course of infection in each group (SIVE or SIVnoE) of animals. (A) Total number of sequences (*y* axis) collected from each tissue at each discretized time point interval (*x* axis) for SIVE (left) and SIVnoE (right) animals. (B) Number of sequences containing SIVE rules (*y* axis) for each tissue and discretized time point (*x* axis) from SIVE (left) and SIVEnoE (right) animals divided according to rule group. Rule groups were comprised of rules connected via at least one amino acid site (Fig. 3B). Early infection was defined as the first 21 days postinfection, late-stage infection as the last 21 days postinfection prior to necropsy, and chronic infection as the time in between. PBMC, peripheral blood mononuclear cell; BAL fluid, bronchoalveolar lavage fluid.

similarity. Overall, signatures for both HIV (15) and SIV could be categorized as exposed within their respective molecular structures (Fig. 7A), with site 216 (rule 4_02) being partially exposed in SIV. The HIV PDB structure held contextual information regarding 2-acetamido-2-deoxy-beta-D-glucopyranose (NAG) residues, revealing information

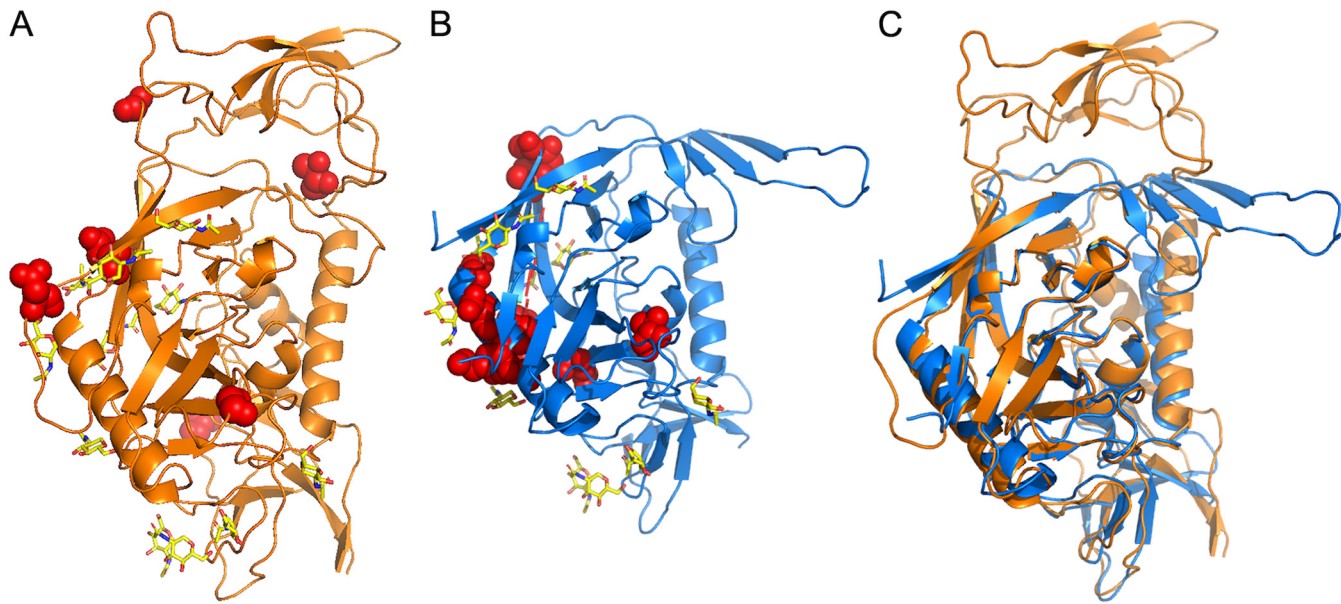

**FIG 7** Amino acid signatures mapped to HIV and SIV Gp120 3-dimensional protein structures. (A) SIV envelope crystal structure (PDB 3JCC). (B) HIV envelope crystal structure (PDB 2NY3). (C) Overlap of SIV and HIV envelope crystal structures from panels A and B. Red spheres represent positions found in SIV and HIV amino acid signatures.

regarding putative aminoglycan contacts among amino acid residues. In HIV, four of the five rules determined by Holman and Gabuzda (15) contained signature sites (5) considered in contact range with NAG, with a distance of ≤4.5 Å considered evidence of putative contact dependency (Table S3, Fig. 7B). For SIV, position 344 (332 according to PDB numbering), predominant among the rules and representing the V3 region boundary, was similarly considered within contact distance of an NAG. Signature site 425 (412 according to PDB numbering), belonging to rule group 5 and found within the V4 region, was also within contact distance. Neighboring V4 signature sites 417 and 420 were considered inserted sites relative to the reference linear sequence and protein structure; consequently, their potential for NAG interaction could not be determined. The V5 signature site 472, belonging to rule groups 2, 4, and 6, was considered to be in relatively close proximity (10.7 Å) to NAG, but not within the considered 4.5 Å distance. It is important to note that SIV NAG contacts were inferred from the HIV PDB (i.e., based on the superimposition of the HIV and SIV Gp120 structures; Fig. 7C), as the resolution for the cryoelectron microscopy SIV Gp120 structure was insufficient for NAG placement. Subtle differences in the HIV and SIV Gp120 structures may translate to differences in NAG placement, resulting in an underestimation of the number of true NAG contacts.

**Phyloanatomic inference of the source of neurovirulent virus in the brain.** While a shared macrophage-tropic phenotype between virus in the CNS and lungs is the simplest explanation for SIVE classification of both tissues, another explanation is that the source of SIVE signatures in the brain is the actual migration of infected alveolar macrophages (AM) from the lungs to the CNS. Epidemiological modeling within a phylogenetic framework within a host, referred to as phyloanatomy (20), offers an *in silico* solution to understanding both viral evolution and dissemination of virus among various anatomical compartments. Derived from the similar framework used to study regional and global migration of pathogens during an epidemic, phyloanatomic analysis assumes a network of isolated tissues tied together via the vascular system and circulating immune cells carrying the virus. Using serial sampling during the course of infection, and assuming a clock-like model of evolution, this method is also able to resolve the timing of relevant evolutionary and epidemiological events, such as the question of whether early viral entry followed by isolated replication in the CNS or late entry of a neuroadapted viral variant that emerged in the periphery is responsible for neurovirulence.

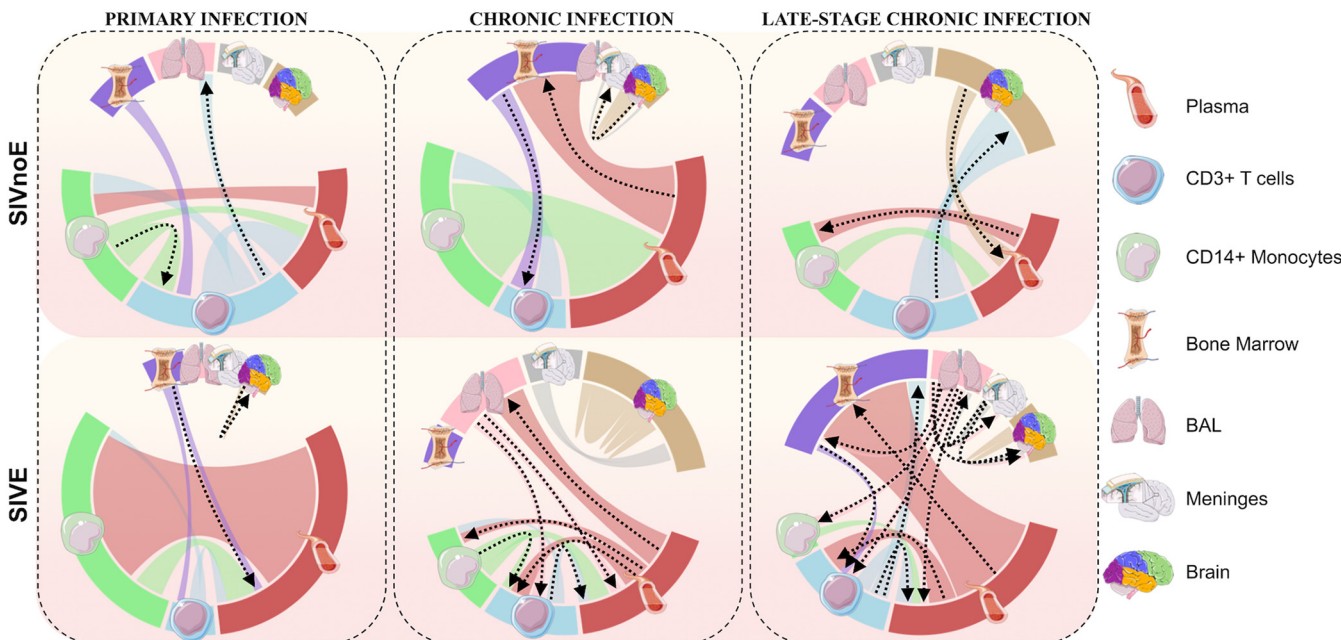

**FIG 8** Graphical representation of viral dispersion patterns over the course of infection in the context of SIVE for the SIV-infected macaque model of HIV infection and neuroAIDS. Bayesian stochastic search variable selection (BSSVS), assuming asymmetric diffusion among discrete anatomical compartments, was implemented in BEAST v1.8.3. The hierarchical phylogenetic model was used to infer spatiotemporal trends in dispersion across animals with ($n = 4$) and without ($n = 3$) SIVE. Diffusion model parameters were allowed to differ between designated time intervals corresponding to early infection (21 days postinfection), AIDS onset (21 days prior to necropsy), and asymptomatic infection (time span between early infection and AIDS onset). Arrows indicate directionality of significant diffusion (Bayes factor >3) and are colored according to tissue of origin.

Previous SIV sequence analyses performed by our group have indicated a potential role for multiple tissues in seeding the brain (45) and a prominent role for non-CNS tissues in the evolution of a neuroadapted virus that enters the brain during the late stage of infection (7). The specific tissue(s) responsible for entry of virus into the brain during early and end-stage disease have not yet been identified in the context of neuropathology but are readily discernible using statistical phyloanatomy techniques (46, 47). Complementing the identification of amino acid signatures of virus capable of replicating in the brain, inference of the viral dissemination patterns among the plethora of infected compartments within the encephalitic host is a critical step toward predicting the risk for development of cognitive impairment. By employing the Bayesian phyloanatomy framework (20), we investigated potential source(s) of CNS virus in the two animal cohorts described in the ML application above, revealing insight into differences in the source and timing of entry of virus between animals that develop SIVE and those that do not.

Bayesian analysis of viral diffusion over the course of infection in SIVE and SIVnoE animals revealed that the virus may be more readily accessible to certain tissues depending on the duration of infection. In both animal groups, viral dissemination during the early phase of infection (first 21 days postinfection [dpi]) was primarily driven by exchange between monocytes, T cells, and blood, as well as from bone marrow to the peripheral T cells, with few exceptions (Fig. 8). One notable exception was the early circulation of virus among the individual brain cortices, or lobes, of the SIVE animals, suggesting earlier brain infection in these animals than in animals without SIVE (SIVnoE). As with all dispersion pathways depicted in Fig. 7 and discussed here, this circulation was considered significant ($BF >3$), characterized by flow of virus from the parietal to the frontal cortex, although the significant anatomical source could not be identified. Following the period of early infection, viral dispersion patterns in the periphery diverged, differing from early infection for both animal groups. During the designated asymptomatic time interval, SIVnoE animal sequences exhibited limited dissemination between, as well as to, the peripheral blood mononuclear cell (PBMC) population and

plasma. Significant exchange between PBMCs and plasma continued into asymptomatic infection, however, for SIVE animals, with the additional contribution of virus from lung AMs. Viral dissemination among the three parenchymal cortices and meninges was detected during asymptomatic infection for both groups, with no significant identifiable anatomical origin.

AIDS-related neuropathogenesis in SIV-infected macaques is characterized by distinct viral dispersion patterns during the late stage of infection. As dispersion patterns continued to diverge from the early infection period and between the SIVE and SIVnoE macaque groups, the onset of AIDS was marked by notable differences in not only the origin of brain viral sequences but of connectivity between peripheral tissues. A loss of significant exchange of virus between peripheral monocytes and T cells was detected for both groups during this time interval compared with earlier infection. It is important to note that as CNS sequences were only available upon necropsy, the exchange of CNS viral populations with those of peripheral tissues may be underestimated for earlier time points. However, nearly uniform sampling over time for the remaining tissues and cell populations offers greater confidence in the indication of a highly dynamic viral population network over the course of SIV infection.

Despite similarly limited contribution of peripheral tissues and cell types to infection of the T cell population, an increased number of dispersion pathways originating from peripheral T cells was observed for the SIVnoE animals compared to early infection. Alternatively, for the SIVE animals, a larger transmission network was observed compared to earlier time periods. These transitions consisted largely of movement into the bone marrow, PBMCs, and peripheral blood, almost exclusively from the lung AMs. This family of macrophages was also the exclusive contributor among sampled locations to cranial virus in SIVE animals, with this exclusion reserved for peripheral T cells in SIVnoE animals, both of which were only significant during late infection.

## DISCUSSION

In this study, we expanded existing ML methods (15, 19) for application to the animal model of HIV-associated neuropathology to identify genetic signatures in the envelope glycoprotein (Gp120) correlated with the onset of neuropathology. We identified 15 amino acid signature sets associated with the presence of SIVE from CNS sequences obtained at necropsy and utilized these signature sets to predict SIVE relevance for sequences amplified from plasma, lymph node, bone marrow, bronchoalveolar lavage (BAL) fluid, and peripheral blood monocytes and T cells, each sampled at different time points during the course of infection and/or treatment. Our goals were to (i) assess the translational significance of such signatures in PLWH to the macaque model, (ii) relate common signatures to protein structure and function, and (iii) to determine if neurovirulent viral variants could be identified outside the spatiotemporal domain of the CNS at the time of symptom onset for more clinically relevant prediction of risk.

Amino acid residues identified in signatures predictive of SIVE using our model were not localized to any subregion of the Gp120 protein, consistent with previous ML studies (15, 19) and acting to explain the seemingly contradictory findings of numerous studies attempting to identify minimal signatures of neurovirulence (12, 48–50). Regardless of nonlocalized signals, residues within the identified SIVE signatures tended to occur within hypervariable regions. Hypervariable regions containing signature sites were expanded beyond V3 to include regions V4 and V5, which were not analyzed in the previously mentioned ML studies. Relatively high variability is often associated with positive selection pressure, as change in these regions can promote immune escape and shifts in cellular tropism (51). Whereas elevated mutability and significant positive selection were observed for several signature sites, these metrics could not be used alone or in conjunction to predict SIVE status and could not fully explain feature importance in the model. Dehghani et al. (52) similarly observed genetic variation within V3 associated with the distinction of brain virus from peripheral blood, representing one of the earlier studies presenting evidence in favor of a potential role for viral evolution in neurological disease progression.

Dehghani et al. (52) specifically noted increased variability for residues 340 and 348 (336 and 344 in SIVmac251 reference and observed among our signatures), proposing affected binding affinity of Gp120 for the CD4 cellular receptor. Differential binding of this receptor is required for more effective replication within cells with lower abundance of surface CD4 molecules, such as monocytes/macrophages (53), which comprise most of the infected population within brain tissue (54). Indeed, replacement of valine with glutamic acid at position 336 was observed in the majority (86%) of SIV sequences extracted during late-stage infection from SIVE animals, as well as a replacement of arginine with tryptophan at 344 in 84% of sequences, including brain, bone marrow, and lymph node samples from the Rhesus macaques used in this study. These findings corroborate the validity of our model and its translation to HIV infection, encouraging increased efforts to expand on the repertoire of residues involved and/or interacting with this region (V3) in the development of neurovirulence.

Ogishi and Yotsuyanagi (19) similarly expanded on the Holman and Gabuzda (15) study, incorporating a differing validation approach in their ML model and more comprehensive data set. While Ogishi and Yotsuyanagi applied a hold-out validation in which they sought to correct the leave-one-sequence-out generalization error from the Holman and Gabuzda study, our cross-validation was explicitly designed to assess the generalization capability of the model. By using a test fold constituted by all the sequences from a single animal, we took a conservative stance, as virus populations from the same animal evolve together and, therefore, cannot be considered independent. With our design, we avoided the risk of our estimates to be overly optimistic, as they would have been, for example, with a leave-one-(sequence)-out or traditional cross-validation that does not consider the sequence origin (i.e., sequences from the same animal are present in both test and training sets). To increase the robustness of the model, Ogishi and Yotsuyanagi (19) used all sequences available from each diagnosed individual within the Los Alamos National HIV Database (55), including blood, lymphatic tissue, and others. As demonstrated in our study, however, disease signatures represent a fraction of the total number of sequences in an individual and may be present in other tissues. Moreover, several studies have demonstrated compartmentalization of sequences in non-CNS tissues (41, 56–60) or even cell types (61, 62), suggesting the introduction of a significant amount of noise with the inclusion of multiple tissues and cell types.

Though phylogenetically distinct from the CNS, monocytes are capable of transmigration across the blood-brain barrier (BBB) and susceptible to viral infection, and are thus believed to play a key role in facilitating the chain of events that result in neurocognitive impairment (54). The expansion of the mature ($CD14^+CD16^+$) monocyte subpopulation from 5 to 10% of all peripheral blood monocytes to ~40% in HIV-infected individuals is predictive of HIV-associated neurocognitive decline and may increase the likelihood of these events (63, 64). While monocytes may act as Trojan horses in the delivery of virus, they may not act alone. The high prevalence of SIVE signature sets observed in this study in lung macrophages, and statistical phyloanatomic evidence of these cells as ancestors of viral progeny in the brain, suggests a supplementary source of brain infection while still consistent with previously described association of neuropathology with a macrophage-tropic phenotype. Infecting viral strain SIVmac239 was reported to replicate poorly in primary AMs derived from BAL fluid samples of healthy virus-negative monkeys, but extremely well at necropsy (168 dpi), indicating this phenotype evolves over the course of infection (65). Similarly, whereas few BAL fluid sequences were classified as SIVE during initial infection in this study, sequences classified as SIVE grew in number during chronic and/or late-stage infection. The presence of SIVE signatures as early as 21 days postinfection, during which the viral population is difficult to discern genetically from the original infecting swarm, suggests that minor variants within the original infecting population may be capable of causing disease but are less fit than the remaining population. Following the emergence of this phenotype in deep tissues such as the lungs, recruitment of proinflammatory macrophages and lymphatic connection to the CNS may result in increased flux of virus.

HIV isolates capable of efficient replication in the brain, deemed "neurotropic," exhibit increased sensitivity to neutralizing antibodies, revealing the trade-off between cellular tropism and immune evasion (11) and providing an explanation for the low fitness of the macrophage-tropic phenotype during early infection. Hypervariable regions within Gp120 have been implicated in this trade-off (66, 67), consistent with our SIVE signature locations. Whereas identity, polarity, and molecular size were demonstrated to be important features at these locations, a common denominator when taking three-dimensional structure into account across HIV signatures of disease were interactions with glycan residues. The most prominent signature site identified in this study (344) was also positioned within contact distance and may not be the only signature with this feature, as insertions relative to the reference protein structure (particularly in V4) inhibited further investigation of glycan interaction potential at these sites. Specifically, asparagine(N)-linked glycosylation of HIV Gp120 has been proposed as a significant mechanism for minimizing immune recognition (68–70), but glycosylated sites have also presented patterns suggestive of selective advantages for replication in the brain, particularly in the V1 and V4 regions (59). Brese and Gonzalez (59) reported that the number and location of N-site patterns were much more conserved in the brain than in remaining lymphoid-derived tissues, suggesting a highly selective microenvironment and functional role for glycosylation that may be distinct from immune evasion (59). A previous comparison of viruses from different time points, including approximate time of transmission, reported less frequent glycosylation during the transmission of HIV type 1 subtypes A and C, suggesting less glycosylated strains are preferably transmitted (68).

We recognize our study has limitations, as do many others diving into diagnosis prediction using ML. The number of animals incorporated into this study was fairly small, and animals undergoing treatment were not represented among SIVE-diagnosed animals, which may harbor differing signatures owing to treatment-mediated selection pressure. Second, whereas macrophage tropism has been implicated as being necessary for neurological infection resulting in disease, there may be more than one evolutionary step required for sufficient neurotropism and/or neurovirulence, which cannot be distinguished in this study. Significantly lower levels of viral diversity and divergence have been observed for macaques associated with disease compared to without disease, indicating separate mechanisms for entry into the brain and efficient replication in the brain (7). Signatures not associated with disease in animals with low-level virus observed in the brain (SIVnoE) do not automatically translate to entry related patterns; they may, in fact, be unrelated to both entry and replication in the brain.

It is also important to note that not all macaque models of CNS disease included in this study were equally represented. Previous passage of virus isolated from the brain of macaques with SIVE has resulted in multiple viral clones that, upon reinfection, lead to rapid and/or high-frequency SIVE in commonly used models of CNS disease (71, 72). Infection of macaques with neurovirulent clone SIVsm804E-CL757 were included in this study and represent high-frequency but slower progression to SIVE (23) relative to other neurovirulent strains (73–75) and immune models (76). These animals were underrepresented (<20%) relative to nonneurovirulent SIVmac251-infected macaques, which consisted of one cohort representing natural progression (low-frequency SIVE) and another rapid, high-frequency progression through CD8$^+$ lymphocyte depletion. Despite underrepresentation of this group within the training data, inclusion of this cohort in three of the six rules, indicating the described model is capable of generalizing across multiple, potentially distinct, models of SIVE. With increasing sequencing endeavors, a larger study with increased representation of this minority group and other animal models of HAND would aid in enhanced determination of the generalizability of the set of rules identified in this study. The benefit of the continued use of the macaque model over application of similar machine learning approaches in PLWH is that the infecting virus is known and can contribute to the identification of potentially multiple signature sets, representative of multiple evolutionary pathways and mechanisms involved in neurovirulence across different models.

Lastly, coverage of the viral envelope gene was increased in this study relative to past studies, contributing to the identification of additional signatures; however, other proteins have been proposed to play a role in neurovirulence (77–81). These proteins are not typically as well represented in SIV/HIV sequencing studies, owing to reasons, including: (i) the difficulty in achieving reliable whole-genome viral sequences (82–84), (ii) the rapid evolution of the envelope gene and consequent contribution to enhanced phylogenetic signal (85, 86), and (iii) the established role of envelope in cellular tropism (49, 87–89). Thus, the reliance on the signatures identified in this study likely do not represent the full complement of genomic sites required for neuroadaptation.

Overall, we showed that, given a sufficiently informative training data set, machine learning can accurately predict SIV/HIV neuropathology through protein biochemical signatures. Though noninvasive sampling and prediction prior to disease onset may not be achievable using these signatures, the machine learning model and its application to the extensive sample collection from multiple macaque models described herein has demonstrated protein biochemical properties that are likely necessary for neuroadaptation. Our results highlight the contribution of lung macrophages as key players in the emergence of neurovirulent strains linked to neuropathology. Additional genomic coverage through whole-genome sequencing and a larger, more diverse set of animal models would provide deeper understanding of the potentially different evolutionary trajectories and mechanisms associated with entry and replication in the brain during progression to neuroAIDS.

## MATERIALS AND METHODS

**Ethics statement.** With respect to the animal (JA41) not included in previous studies, all animal procedures were performed by the Tulane National Primate Research Center (TNPRC) in accordance with Tulane University's Institutional Animal Care and Use Committee (IACUC) protocol P0376. The macaque cohort to which this animal belonged were housed in groups or pairs at Tulane animal facilities accredited by the Association for Assessment and Accreditation of Laboratory Animal Care (AAALAC). Details surrounding animal welfare, including environmental parameters and standard practices for treatment of nonhuman primates in research, were followed as outlined in the *Guide for the Care and Use of Laboratory Animals* (90). All possible measures were employed to minimize discomfort of the animals. Once IACUC defined endpoints were reached, macaques were humanely euthanized following the standard method of euthanasia for nonhuman primates at the TNPRC, consistent with the recommendations of the American Veterinary Medical Association Guidelines on Euthanasia.

**Study population.** Sequences from a variety of Rhesus macaque models of HIV infection were included in this study to increase the robustness of the machine learning analysis (Table 1). Disease progression was achieved both naturally (7, 23, 25) and through CD8$^+$ lymphocyte depletion (rapid) (7), with one of the following inoculation methods: neurovirulent clone (SIVsm804E-CL757) (23) or nonneurovirulent viral swarm (SIVmac251) (7, 25). SIV-mediated encephalitis (SIVE) was diagnosed in all cohorts at necropsy based on brain pathology, primarily focusing on the presence of SIV-positive multinucleated giant cells and lesion formation with accompanying lymphocytic infiltration. No observable pathogenic lesions or evidence of SIVE were detected in the cohort described in Perez et al. (25) From this cohort, two animals undergoing 24 weeks of cART (Tenofovir + Emtricitabine) with no evidence of SIVE were included in training and validation in this study.

Additional tissues obtained from the cohorts described by Rife et al. (7) were utilized in the determination of prevalence of neurovirulent signatures outside the CNS for SIVE and SIVnoE diagnoses. Peripheral blood mononuclear cells, lymph node tissue, and laminar propria lymphocytes were also obtained at 198 days post-SIVmac251 infection from an additional SIVnoE animal treated with Tenofovir, Emtricitabine, and Raltegravir for approximately 20 weeks prior to treatment interruption (18 days prior to last sample collection). Plasma samples were also obtained from this animal at 14, 21, and 28 dpi (pre-cART) and again at 198 dpi (post-cART), for a total of 72 RNA sequences across all tissues.

**Sequencing and phylogenetic analysis.** A variety of methods were used for the cohorts described to obtain envelope gene sequences from DNA and/or RNA. The majority of sequences were derived using single-genome amplification (7) ($n$ = 4,109 sequences), though consensus sequences obtained from PCR purification (25) or TOPO TA cloning (23) were also included ($n$ = 109 sequences).

A total of 4,218 envelope *gp120* nucleotide sequences were aligned using the Muscle alignment algorithm (91) in AliView v1.26 (92) and translated to amino acid sequences (also in AliView). Using this approach, ambiguous amino acids resulting from ambiguous coding nucleotides were unresolved. Sequences with premature stop codons were removed, so that final amino acid sequences included the Gp120 variable regions V1 to V5 and constant regions C1 to C4. A maximum likelihood phylogenetic tree for all amino acid sequences was then generated in IQ-TREE v2.1.3 (93). Similar to Holman and Gabuzda (15), the Jones-Taylor Thornton (JTT) substitution matrix (94) was used in tree reconstruction, which corrects for multiple, unobserved amino acid substitutions. Base frequencies were determined empirically.

**Weighting and translation of sequences to amino acid properties.** Following the Holman and Gabuzda (15) method, to ensure that sequences that were obtained from different sources were weighted equally, and the number of sequences per animal was balanced, the weight per sequence was calculated by:

$$\frac{total\ number\ of\ sequences\ in\ the\ data\ set}{total\ number\ of\ animals\ in\ that\ data\ set\ \times\ total\ sequences\ in\ an\ animal}$$

Since multiple amino acids can have similar biochemical properties, four numeric factors (or attributes) to describe the functional role of the amino acid in each sequence were included in the analysis: polarity, secondary structure, molecular size, and electrostatic charge. These factors, also used in the Holman and Gabuzda study, were previously derived (27) using multivariate statistical analysis of physiochemical and biological indices reported in the online AAindex database (https://www.genome.jp/dbget/aaindex.html).

**Model training and validation.** We utilized sequences from Rhesus macaques infected with SIVmac251 (7) as well as sequences from external data (23, 25). One obstacle encountered during data set assembly when working with brain-derived sequences is sufficient sample size. We, therefore, included CNS sequences from other research groups with similarly diagnosed cohorts (23, 25). A total of 553 sequences from 15 Rhesus macaques (Table 1) were used to train the model, 445 of which were provided from in-house data (7) and 109 from external sources (23, 25). All sequences included variable regions 1 to 5 and constant regions C1-C4 of the Gp120 region.

Feature selection from Weka (v3.8.5) was implemented on all sequences in order to use relevant features from the original set of features and to increase the performance of the model (95). Sites corresponding to deletions relative to the alignment or ambiguous amino acids were treated as missing data ("?") for all features. We used the J48 wrapper based on information gain to extract the most relevant features (96), with WrapperSubsetEval and the BestFirst greedy hill-climbing algorithm (97), both with default parameters.

Following feature selection, the model was trained using the Weka projective adaptive resonance theory (PART) rule-learning algorithm with default parameters using R (v4.1.1) (98) to classify sequences by SIVE/SIVnoE diagnosis. PART is based on the C4.5 algorithm, a decision tree generating a hierarchical set of rules for classification (99). PART outputs an interpretable and ordered set of rules (SIVE and SIVnoE). Each rule includes amino acid requirements with which a sequence needs to comply to be classified. Since diagnosis is determined at necropsy, we generated the PART model using tissue sections from the brain, meninges, and spinal cord taken at necropsy. This model resulted in a total of 15 hierarchical rules for classification. The individual signatures obtained within the set of rules were interpreted as amino acid signatures. The last rule applies a default classification, if no other rule successfully matches the amino acid sequence (100).

Leave-one-animal-out cross validation was used to validate the trained model, sequentially holding all the sequences from one animal out from the training set, training the classifier, and evaluating the prediction for the left-out animal's class. The leave-one-out animal approach was applied, as opposed to leave-one-sequence-out, considering that diagnosis (SIVE or SIVnoE) is assigned at the level of the subject and not at the level of individual sequences; furthermore, the genetic relatedness of the sequences derived from the same subject can lead to biased classification (19). Animals were classified as SIVE when most of their constituent sequences followed at least one rule predictive of SIVE. One hundred percent was not used a cutoff here owing to the natural sequence heterogeneity present in HIV/SIV-infected tissues. Additionally, given that each rule was comprised of more than one signature site, neuroadaptation was not assumed to be defined as a binary, or all-or-nothing, trait—i.e., all signatures may not emerge simultaneously and may not be necessary for viral entry and/or replication in the brain, though they may collectively provide a significant selective advantage, lending to their presence as a majority of the virus population in the CNS. The performance of the model (e.g., accuracy, precision) was calculated only on test data.

**Estimation of site-specific mutational entropy.** Shannon entropy was used as a metric of genetic diversity for each site across the population of sequences and was calculated using the following:

$$SE(X) = -\sum_{i=1}^{N=20} p(x_i)log_2p(x_i)$$

Where $p(x_i)$ is the probability that site $X$ will contain amino acid $x_i$ among all possible amino acids ($x_1$, $x_2$, $x_3$, . . ., $x_N$). Gaps and ambiguous characters were ignored.

One-way ANOVA with Tukey's multiple comparisons *post hoc* test was used to determine if differences in entropy between classifications of sequences (i.e., signature and nonsignature sites, training, and application) were significant.

**Bayesian graphical model of amino acid coevolution.** The Bayesian graphical model (BGM) implemented in Datamonkey (101, 102) (http://www.datamonkey.org/) was used to determine the probability of codependence of amino acid change within the sequence alignment with default parameters.

**Identification of selection pressure.** The relative rate ratios of nonsynonymous (*dN*) and synonymous (*dS*) codon substitutions were calculated for each animal-specific nucleotide alignment using the mixed effects model of evolution (MEME) (32), available on the Datamonkey Adaptive Evolution Server (102) (http://www.datamonkey.org/). The best-fit codon substitution model (MG94) was used, as determined in Datamonkey. The default threshold (*P* value <0.10) was used to determine evidence of significant selection.

The percentage of branches within the corresponding phylogenetic tree containing evidence of positive selection for each signature site was used to further evaluate the pervasiveness of selection for each animal.

The fast, unconstrained Bayesian approximation method of selection (33) was also used to determine if signature sites were considered to experience purifying selection in animals infected with neuro-virulent clone SIVsm804E-CL757 and only sampled at necropsy.

**Gp120 protein structural analysis.** Crystal structures of HIV (PDB 2NY3) (43) and SIV (PDB 3JCC) (44) Gp120 were used for mapping individual signature sites. Secondary-structure matching (SSM) in Crystallographic Object-Oriented Toolkit (COOT) (103) was used to superimpose SIV and HIV Gp120 structures, using 2NY3 as the reference structure. Interatomic distances between amino acid and 2-acetamido-2-deoxy-beta-D-glucopyranose (NAG) residues were measured in COOT. PyMOL (https://pymol.org/2/) (104) was used to generate molecular graphic images.

**Bayesian phyloanatomy.** Bayesian genealogical tree reconstruction for individual macaque-specific gp120 sequence alignments was performed using BEAST v1.8.3 (105, 106) (available from http://beast.bio.ed.ac.uk/), assuming an uncorrelated relaxed molecular clock model of evolutionary rate variation across branches (107) and Bayesian Skyride demographic model (108, 109). Prior information can be observed from the representative xml found in https://github.com/rifebd88/SIV_Phyloanatomy. A subset (500) of systematically drawn trees from the resulting posterior tree distribution for each macaque was obtained for use as empirical data for further analysis, owing to the computational complexity of integrating over the possible tree space for many taxa (50). The tree subsets and script used to generate them can also be found at https://github.com/rifebd88/SIV_Phyloanatomy. Macaque-specific gp120 sequence alignments were categorized according to SIVE diagnosis (SIVE and SIVnoE) and treated as individual partitions according to a hierarchical phylogenetic model (HPM) in subsequent Bayesian analysis. An asymmetric transition rate matrix within the Bayesian stochastic search variable selection (BSSVS) allowed for inferred directionality of viral dissemination patterns occurring at significantly nonzero rates between discrete sampled anatomical compartments. Within the HPM, an epoch model (110) was used to infer spatiotemporal dissemination patterns during intervals of time over the duration of the evolutionary history of the viral population (assumed constant over time). Time intervals included the first 21 dpi (approximating acute infection), last 21 days prior to necropsy (approximating AIDS onset), and the duration between (approximating asymptomatic infection). Prior information can be observed from the representative xml found in https://github.com/rifebd88/SIV_Phyloanatomy. Effective Markov chain Monte Carlo sampling (111) for all Bayesian analyses was assessed by calculating the effective sample size (ESS) for each estimated parameter. ESS values > 200, calculated in Tracer (available from http://beast.bio.ed.ac.uk/Tracer), were considered suitable indicators of effective sampling. Bayes factor (BF) support (BF > 3) (112) for nonzero transition rates of SIV viral lineages among discrete anatomical locations within the host was assessed at the hierarchical level using SPREAD (113) (latitude and longitude coordinate designations of "1" for all anatomical locations).

**Figure generation.** Representative feature selection based on biochemical properties (Fig. 3B) was generated in biorender (114). All remaining figures were generated in R (98) using the following packages: ggplot (115), dplyr (116), and seqinr (117).

**Data availability.** All of the sequences used in this study have been described previously (7, 30, 45, 118) and are accessible in GenBank (accession numbers JF765272 to JF766081 [Mac251-DEP], KR999328 to KR999727 [Mac251-NP N02 and N10], and KX081254 to KX081353; KX081479 to KX081498; KX081619 to KX081702; KX081840 to KX081862; KX082029 to KX082107; KX082229 to KX082252; KX082428 to KX082531 [Mac251-NP N09], and MG931034 to MG931480 [N03]; ON714061 to ON714132 [JA41]; SIVE external MF370654 to MF370782; SIVnoE external MF284715 to MF284792).

## SUPPLEMENTAL MATERIAL

Supplemental material is available online only.

**SUPPLEMENTAL FILE 1**, PDF file, 0.4 MB.

## ACKNOWLEDGMENTS

The authors acknowledge the Stephany W. Holloway University Chair in AIDS Research and the University of Florida Research Computing (http://researchcomputing.ufl.edu) for providing computational resources and support that have contributed to the research results reported in this publication. This work was supported by National Institutes of Health, National Institute of Neurological Disease and Stroke (NINDS, R01NS063897), and the University of Florida MacClamma Research Fellowship Fund.

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
