## [Reviewer comments · Microbiology Spectrum]

Microbiology Spectrum

Machine learning prediction and phyloanatomic modeling of viral neuroadaptive signatures in the macaque model of HIV-mediated neuropathology

Andrea Ramirez-Mata, David Ostrov, Marco Salemi, Simone Marini, and Brittany Rife Magalis

Corresponding Author(s): Brittany Rife Magalis, University of Florida Emerging Pathogens Institute

Review Timeline:

Submission Date:	August 16, 2022
Editorial Decision:	October 19, 2022
Revision Received:	January 24, 2023
Accepted:	February 6, 2023

Editor: Yongjun Sui

Reviewer(s): Disclosure of reviewer identity is with reference to reviewer comments included in decision letter(s). The following individuals involved in review of your submission have agreed to reveal their identity: Syed Hani Hassan Abidi (Reviewer #1)

Transaction Report:

DOI: <https://doi.org/10.1128/spectrum.03086-22>

October 19, 2022

Dr. Brittany Rife Magalis
University of Florida Emerging Pathogens Institute
Department of Pathology, Immunology, and Laboratory Medicine
2055 Mowry Rd
Gainesville, FL 32610

Re: Spectrum03086-22 (Machine learning prediction and phyloanatomic modeling of viral neuroadaptive signatures in the macaque model of HIV-mediated neuropathology)

Dear Dr. Brittany Rife Magalis:

Link Not Available

Sincerely,

Yongjun Sui

Journals Department
Reviewer comments:

Reviewer #1 (Comments for the Author):

In this study, Magalis et al used ML and phylogenetic modelling approaches to identify amino acid signatures of SIVE (serving as a model for HAND). I found the study to be interesting and agree with most of their findings. However, there were many places where sufficient details given below:

Introduction

- Line 53-56: In addition to viral or immune mechanisms, do mention the role of antiretroviral drugs in HAND.

Methods

- How were the sequences obtained/generated from each animal? What was the sequence length? How many sequences were obtained? Were these population or clonal sequences? Were they converted to consensus? Describe details
- How were DNA sequences converted to amino acids? Were there any sequence ambiguities and/or variabilities observed at nucleotide levels, and how were these resolved?
- Why was JTT model for phylogeny selected? What were the additional assumptions for trees: such as site selection, rate of evolution, etc?
- Line 514: simply giving a reference is not sufficient. Please describe how each property was calculated.
- Line 515-521: How many were from this study and how many sequences were from other studies Was the length equal? Were these sequences representative of the current study, in terms of homology and coverage?
- Was selection pressure calculated? It would be interesting to see if these amino acids were under positive, negative or neutral selection.
- While I agree with the leave-one-animal-out approach, by not adopting the leave-one-sequence-out, how did the authors take care of within-host sequence heterogeneity and its relationship to the diseases?
- Details of sequence extraction from each tissue need to be added.
- Line 583, 584, 596, 606 : There is a '?'. Please check
- What was the sensitivity of the ML model considering the small size?
- Please check if animal study review committee approval is given in the paper

Results

- Line 135: For how many sequences the time-point information was available?
- Fig 2: It is very hard to determine the clustering. Authors should give the rectangular tree here or in supplementary that can be used to see the clustering.
- What results (table or figure) show time-point information? I could not find the supporting data to see the differences between early and late variants. And how do they cluster?
- Fig 3. Grey is hardly visible. Change to some other colour.
- I think it would be useful to have a representation of amino acid identity, polarity, molecular size, and role in protein secondary structure, electrostatic charge of important and not-important amino acids/sites. This can be an alignment figure with all attributes identified.
- Try to remove as many references from results as possible. I understand some are important, but at place, the results looked like discussion.

Staff Comments:

Preparing Revision Guidelines

Please return the manuscript within 60 days; if you cannot complete the modification within this time period, please contact me. If you do not wish to modify the manuscript and prefer to submit it to another journal, please notify me of your decision immediately so that the manuscript may be formally withdrawn from consideration by Microbiology Spectrum.

We are grateful for the comments and suggestions provided by the reviewers and have addressed them in the following document, with responses in blue and replaced text in italic. Accompanying revisions to the original text can also be found in blue in the newly submitted manuscript.

Reviewer #1 (Comments for the Author):

In this study, Magalis et al used ML and phylogenetic modelling approaches to identify amino acid signatures of SIVE (serving as a model for HAND). I found the study to be interesting and agree with most of their findings. However, there were many places where sufficient details given below:

Introduction

- Line 53-56: In addition to viral or immune mechanisms, do mention the role of antiretroviral drugs in HAND.

The introduction (lines 55-59) has been revised to include potential drug-mediated mechanisms, which are also discussed in the existing cited references, as follows:

“Though more commonly observed among PLWH that have progressed to end-stage acquired immunodeficiency syndrome (AIDS) (Gelman, 2012), not all PLWH (approximately 25%) are diagnosed (Xu, 2017), suggesting a distinct viral-, immune-, or potentially drug-mediated mechanism associated with disease pathogenesis that has yet to be identified (Nightingale, 2014; Walsh & Reinke, 2014).”

Methods

- How were the sequences obtained/generated from each animal? What was the sequence length? How many sequences were obtained? Were these population or clonal sequences? Were they converted to consensus? Describe details

Table 1 has been modified to include rows for virus inoculum composition, animal species, animal model, and sequence origination method for training/validation and application datasets. Total sequence number is already included. As sequences in this study have already been described in other publications (also listed in Table 1), more details on the methods can be found in these publications. In addition, the detailed section in the Materials and Methods, largely focused on our group’s cohort (already published), was replaced with a summary describing all cohorts in the study (lines 561-586), as follows:

“Study population. Sequences from a variety of Rhesus macaque models of HIV infection were included in this study to increase the robustness of the machine learning analysis (Table 1). Disease progression was achieved both naturally (Matsuda, 2017; Perez, 2017; Rife, 2016) and through CD8+ lymphocyte depletion (rapid) (Rife, 2016), with one of the following inoculation methods: neurovirulent clone (SIVsm804E-CL757) (Matsuda, 2017) or non-neurovirulent viral swarm (SIVmac251) (Rife, 2016; Perez, 2017). SIV-mediated encephalitis (SIVE) was diagnosed in all cohorts at necropsy based on brain pathology, primarily focusing on the presence of SIV-positive multinucleated giant cells and lesion formation with accompanying lymphocytic infiltration. No observable pathogenic lesions or evidence of SIVE were detected in the cohort described in Perez et al. From this cohort, two animals undergoing 24 weeks of cART (Tenofovir + Emtricitabine) with no evidence of SIVE were included in training and validation in this study.

Additional tissues obtained from the cohorts described by Rife et al. were utilized in the determination of prevalence of neurovirulent signatures outside the CNS for SIVE and SIVnoE diagnoses. Peripheral blood mononuclear cells, lymph node tissue, and lamina propria lymphocytes were also obtained at 198 days post-SIVmac251 infection from an additional SIVnoE animal treated with Tenofovir, Emtricitabine, and Raltegravir for approximately 20 weeks prior to treatment interruption (18 days prior to last sample collection). Plasma samples were also obtained from this animal at 14, 21, and 28 dpi (pre-cART) and again at 198 dpi (post-cART), for a total of 72 RNA sequences across all tissues.

Sequencing and phylogenetic analysis. *A variety of methods were used for the cohorts described to obtain envelope gene sequences from viral DNA and/or RNA. The majority of sequences were derived using single-genome amplification (Rife, 2016) (n=4109 sequences), though consensus sequences obtained from PCR purification (Perez, 2017) or TOPO TA cloning (Matsuda, 2017) were also included (n=109 sequences)."*

The results section (lines 113-116) has also been expanded to emphasize the variety of inclusion, as follows:

"Viral Gp120 sequences used in model training were extracted from the CNS of animals originating from multiple cohorts comprised of different macaque models of disease progression, (Table 1), including infection with a neurovirulent strain (Matsuda, 2017) and CD8+ lymphocyte depletion (Rife, 2016). Each animal was histopathologically diagnosed at necropsy as with (SIVE) or without SIVE (SIVnoE)."

- How were DNA sequences converted to amino acids? Were there any sequence ambiguities and/or variabilities observed at nucleotide levels, and how were these resolved?

A description of amino acid translation has been added to the Materials and Methods (lines 587-596) as follows:

"A total of 4218 envelope gp120 nucleotide sequences were aligned using the Muscle alignment algorithm (90) in AliView v1.26 (91) and translated to amino acid sequences (also in AliView). Using this approach, ambiguous amino acids as a result of ambiguous coding nucleotides were unresolved. Sequences with premature stop codons were removed, so that final amino acid sequences included the Gp120 variable regions V1 to V5 and constant regions C1 to C4."

Furthermore, we discovered that ambiguities in the form of gaps were being treated incorrectly, resulting in the loss of signature sites and, consequently, rules. These additional signature sites increased both the specificity and sensitivity of the analysis without changing the overall story, as they were still prevalent among lung macrophage sequences and could not be evaluated in the protein structure. Methods have been modified to clarify the treatment of gaps as missing data and results modified in several places throughout the text to include these additional sites.

- Why was JTT model for phylogeny selected? What were the additional assumptions for trees: such as site selection, rate of evolution, etc?

The JTT model was chosen so as to replicate more closely the original Holman and Gabuzda study using HIV sequences, as the primary goal of this study was to determine the

translation of the results to the SIV-macaque model. This model is commonly used in HIV phylogenetic analysis as it is able to faithfully correct for multiple substitutions and performs well among non-HIV-specific models (Nickle, 2007). This information has now been summarized in the Methods section (lines 592-596) as follows:

“A maximum likelihood phylogenetic tree for all amino acid sequences was then generated in IQ-TREE v2.1.3 (Minh, 2020). Similar to Holman and Gabuzda, the Jones-Taylor Thornton (JTT) substitution matrix (Jones, 1992) was used in tree reconstruction, which corrects for multiple, unobserved amino acid substitutions. Base frequencies were determined empirically.”

- Line 514: simply giving a reference is not sufficient. Please describe how each property was calculated.

This line (now lines 602-604) has been revised as follows:

“These factors, also used in the Holman and Gabuzda study, were previously derived (Atchley, 2005) using multivariate statistical analysis of physiochemical and biological indices reported in the online AAindex database of properties (<https://www.genome.jp/dbget/aaindex.html>).”

- Line 515-521: How many were from this study and how many sequences were from other studies Was the length equal? Were these sequences representative of the current study, in terms of homology and coverage?

Lines 610-612 (Methods) have been modified and Table 1 added to describe the cohort contributions and emphasize the differences in representation as follows:

“A total of 553 sequences from 15 Rhesus macaques (Table 1) were used to train the model, 444 of which were provided from in-house data (Rife, 2016) and 109 from external sources (Matsuda, 2017; Perez, 2017). All sequences included covered variable regions 1-5 and constant regions C1-C4 of the envelope glycoprotein (Gp120) region.”

The external sequences were considered representatively adequate for this study owing to gene coverage and diagnostic method (histopathology), as well as independence and lack of convergent evolution (demonstrated in the phylogeny and discussed in the results). Moreover, the rules applied to all SIVE animals, regardless of cohort. A supplementary table (S1) has been added, breaking down classification of sequences and animals per animal cohort, and the following has been added to the discussion (lines 504-523) to highlight this point:

“It is also important to note that not all macaque models of CNS disease included in this study were equally represented. Previous passage of virus isolated from the brain of macaques with SIVE has resulted in multiple viral clones that, upon re-infection, lead to rapid and/or high-frequency SIVE in commonly used models of CNS disease. Infection of macaques with neurovirulent clone SIVsm804E-CL757 alone, or in combination with non-neurovirulent SIVsmE543-3, were included in this study and represent high-frequency but slower progression to SIVE (Matsuda, 2017) relative to other neurovirulent strains and immune models. These animals were underrepresented (<20%) relative to non-neurovirulent SIVmac251-infected macaques, which consisted of one cohort representing natural progression (low-frequency SIVE) and another rapid, high-frequency progression through CD8+ lymphocyte depletion. Despite under-representation of this group within the training data, inclusion of this cohort in three of the six rules, indicating the described model is capable of generalizing across multiple,

potentially distinct, models of SIVE. With increasing sequencing endeavors, a larger study with increased representation of this minority group and other animal models of HAND would aid in enhanced determination of the generalizability of the set of rules identified in this study. The benefit of the continued use of the macaque model over application of similar machine learning approaches in PLWH is that the infecting virus is known and can contribute to the identification of potentially multiple signature sets, representative of multiple evolutionary pathways and mechanisms involved in neurovirulence across different models.”

• Was selection pressure calculated? It would be interesting to see if these amino acids were under positive, negative or neutral selection.

We have included the results (now Figures 5 and S2) of diversifying and purifying selection analysis, as suggested by the reviewer (lines 208-236):

“Alternative to epistasis, molecular adaptation at the level of an individual residue often requires flexibility in terms of amino acid change, represented as elevated mutational entropy, and can be driven by positive selection. Selection experienced by individual amino acids can be identified, and quantified, through the estimation of underlying relative non-synonymous (dN) and synonymous (dS) codon substitution rates within the nucleotide data. Despite the relatively wide range of mutational entropy between the two signature sites, selection pressure experienced by sites required for neurovirulence may occur only transiently during the course of infection in the midst of dynamic immune responses (29) and target cell distributions (30), resulting in fluctuations in dN over time. We, therefore, applied the mixed effects model of evolution (MEME) to the nucleotide sequence data, designed to detect both pervasive and episodic selection (31). Using this approach, a total of 127 amino acid sites were identified as experiencing significant positive selective pressure ($p < 0.10$). All signature sites predictive of SIVE were considered to have experienced significant positive selection pressure, though not restricted to SIVE animals (Figure 5a). Whereas positive selection of signature sites was considered approximately 2-fold greater than that of non-signature sites, selection for signature sites was only observed on average across 1.8% of branches (Figure S2), indicating selection was largely episodic and potentially explaining why positive selection for these sites was largely not observed among the SIVsm804E-CL757-infected animals sampled only at necropsy (Figure 5b). Of the eight sites, highly prevalent V3 signature site 344 and less prevalent V4 signature site 420 were the only sites observed among both SIVmac251- and SIVsm804E-CL757-infected cohorts, among which site ... was specific to the CD8-depleted animals only and site to both CD8-depleted and non-depleted animals. Assuming signature sites not observed in the SIVsm804E-CL757 cohort could be required to maintain fitness in animals directly infected with a neurovirulent clone, analysis of purifying selection was also undertaken using a fast, unconstrained Bayesian approximation (FUBAR)(32); however, synonymous rate variation was not considered significantly greater ($p < 0.05$) than that of non-synonymous substitutions for these sites in these animals.”

Discussion of the results have been added to lines 402-407:

“Relatively high variability is often associated with positive selection pressure, as change in these regions can promote immune escape and shifts in cellular tropism (50). Whereas elevated mutability and significant positive selection were observed for several

signature sites, these metrics could not be used alone or in conjunction to predict SIVE status and could not fully explain feature importance in the model.”

The following methods section has also been added to lines 662-675:

“The relative rate ratios of non-synonymous (dN) and synonymous (dS) codon substitutions were calculated for each animal-specific nucleotide alignment using the mixed effects model of evolution (MEME) (31), available on the Datamonkey Adaptive Evolution Server (101) (<http://www.datamonkey.org/>). The best-fit codon substitution model (MG94) was used, as determined in Datamonkey. The default threshold (p -value <0.10) was used to determine evidence of significant selection.

The percentage of branches within the corresponding phylogenetic tree containing evidence of positive selection for each signature site was used to further evaluate the pervasiveness of selection for each animal.

The fast, unconstrained Bayesian approximation method of selection (32) was also used to determine if signature sites were considered to experience purifying selection in animals infected with neurovirulent clone SIVsm804E-CL757 and only sampled at necropsy.”

- While I agree with the leave-one-animal-out approach, by not adopting the leave-one-sequence-out, how did the authors take care of within-host sequence heterogeneity and its relationship to the diseases?

As referenced in the paper, previous studies (including by our group) have demonstrated compartmentalization of sequences in the brain – i.e., diversity is significantly reduced in the brain of diseased animals as compared with the brain of non-encephalitic animals, as well as compared with other tissues. Regardless, within-host sequence heterogeneity is taken into consideration throughout the paper – for example, both the % of sequences and % of animals are taken into consideration when evaluating performance, with an animal considered accurately classified if the majority of sequences are correctly classified. One hundred percent is not used here, owing to sequence heterogeneity and the assumption that neuroadaptation is not a binary process -i.e., all signatures may not be necessary for entry and/or replication in the brain, though they may collectively provide a significant selective advantage. These additional details have now been included in the methods (lines 640-647) as reflected below:

“Animals were classified as SIVE when the majority of their constituent sequences followed at least one rule predictive of SIVE. One hundred percent was not used a cutoff here owing to the natural sequence heterogeneity present in HIV/SIV-infected tissues. Additionally, given that each rule was comprised of more than one signature site, neuroadaptation was not assumed to be defined as a binary, or all-or-nothing, trait - i.e., all signatures may not emerge simultaneously and may not be necessary for viral entry and/or replication in the brain, though they may collectively provide a significant selective advantage, lending to their presence as a majority of the virus population in the CNS.”

- Details of sequence extraction from each tissue need to be added.
Please see response above.

- Line 583, 584, 596, 606 : There is a '?'. Please check
The loss of references has been corrected.
- What was the sensitivity of the ML model considering the small size?
The sensitivity is provided (recall) in Table 2, which was 97% on a per sequence level and 86% on a per animal level.
- Please check if animal study review committee approval is given in the paper
Considering one animal was included in this study that has not been previously published, an ethics section was added as follows (lines 548-560):

“Ethics statement. With respect to the animal (JA41) not included in previous studies, all animal procedures were performed by the Tulane National Primate Research Center (TNPRC) in accordance with Tulane University’s Institutional Animal Care and Use Committee (IACUC) protocol P0376. The macaque cohort to which this animal belonged were housed in groups or pairs at Tulane animal facilities accredited by the Association for Assessment and Accreditation of Laboratory Animal Care (AAALAC). Details surrounding animal welfare including environmental parameters and standard practices for treatment of non-human primates in research were followed as outlined in the Guide for the Care and Use of Laboratory Animals (NRCNA, 2011). All possible measures were employed to minimize discomfort of the animals. Once IACUC defined endpoints were reached, macaques were humanely euthanized following the standard method of euthanasia for non-human primates at the TNPRC, consistent with the recommendations of the American Veterinary Medical Association Guidelines on Euthanasia.”

Results

- Line 135: For how many sequences the time-point information was available?
This information is provided in Fig 6a.
- Fig 2: It is very hard to determine the clustering. Authors should give the rectangular tree here or in supplementary that can be used to see the clustering.
Owing to the large amount of data in the tree, clustering is difficult to discern regardless of tree form. We have, therefore, made the tree available in the github repository for those interested in the examining individual clades and have also included insets of the amino acid sequence phylogeny in Figure 2 showing absence of cohort and SIVE status clustering. However, we felt the amino acid sequence phylogeny was not a sufficient representation of animal independence and have thus added an additional subpanel to Figure 2 (b) showing the underlying nucleotide sequence phylogeny and inset demonstrating animal-specific clustering. The following has been added to the results (lines 144-150), describing this new panel:

“Whereas macaque-specific clustering of amino acid sequences was not observed for necropsy samples used in training and validation, corresponding nucleotide sequences clustered significantly (bootstrap support $\geq 90\%$) according to animal (Figure 2b) This finding is consistent with the presence of immune-mediated convergent evolution at the protein level (McClellan, 2013), but is also indicative of host-specific genetic drift of the virus population and, thus, allowing for each animal to be treated independently within the model.”

- What results (table or figure) show time-point information? I could not find the supporting data to see the differences between early and late variants. And how do they cluster?
Please see answers to comments above.

- Fig 3. Grey is hardly visible. Change to some other colour.
The grey has been replaced with blue in this figure and figure legend changed to reflect this.

- I think it would be useful to have a representation of amino acid identity, polarity, molecular size, and role in protein secondary structure, electrostatic charge of important and not-important amino acids/sites. This can be an alignment figure with all attributes identified.
A representative figure panel has been added to Fig 3, as suggested by the reviewer, along with an additional panel demonstrating the hierarchy, or nested nature, of the learned rules.

- Try to remove as many references from results as possible. I understand some are important, but at place, the results looked like discussion.
We have either removed sentences or placed sentences in the Discussion section that we deemed more relevant for actual discussion.

February 6, 2023

Dr. Brittany Rife Magalis
University of Florida Emerging Pathogens Institute
Department of Pathology, Immunology, and Laboratory Medicine
2055 Mowry Rd
Gainesville, FL 32610

Re: Spectrum03086-22R1 (Machine learning prediction and phyloanatomic modeling of viral neuroadaptive signatures in the macaque model of HIV-mediated neuropathology)

Dear Dr. Brittany Rife Magalis:

Your manuscript has been accepted, and I am forwarding it to the ASM Journals Department for publication. You will be notified when your proofs are ready to be viewed.

Sincerely,

Yongjun Sui
Editor, Microbiology Spectrum
